# BERT Lost Patience
# Won't Be Robust to Adversarial Slowdown

**Zachary Coalson, Gabriel Ritter, Rakesh Bobba, and Sanghyun Hong**
Oregon State University
{coalsonz, ritterg, bobbar, hongsa}@oregonstate.edu

## Abstract

In this paper, we systematically evaluate the robustness of multi-exit language models against adversarial slowdown. To *audit* their robustness, we design a slowdown attack that generates natural adversarial text bypassing early-exit points. We use the resulting WAFFLE attack as a vehicle to conduct a comprehensive evaluation of three multi-exit mechanisms with the GLUE benchmark against adversarial slowdown. We then show our attack significantly reduces the computational savings provided by the three methods in both white-box and black-box settings. The more complex a mechanism is, the more vulnerable it is to adversarial slowdown. We also perform a linguistic analysis of the perturbed text inputs, identifying common perturbation patterns that our attack generates, and comparing them with standard adversarial text attacks. Moreover, we show that adversarial training is ineffective in defeating our slowdown attack, but input sanitization with a conversational model, e.g., ChatGPT, can remove perturbations effectively. This result suggests that future work is needed for developing efficient yet robust multi-exit models. Our code is available at: https://github.com/ztcoalson/WAFFLE

## 1 Introduction

A key factor behind the recent advances in natural language processing is the *scale* of language models pre-trained on a large corpus of data. Compared to BERT [5] with 110 million parameters that achieves the GLUE benchmark score [36] of 81% from three years ago, T5 [29] improves the score to 90% with 100× more parameters. However, pre-trained language models with this scale typically require large memory and high computational costs to run inferences, making them challenging in scenarios where latency and computations are limited.

To address this issue, *input-adaptive* multi-exit mechanisms [32, 41, 45, 40, 42, 48, 21, 44] have been proposed. By attaching internal classifiers (or early exits) to each intermediate layer of a pre-trained language model, the resulting multi-exit language model utilizes these exits to stop its forward pass preemptively, when the model is confident about its prediction at any exit point. This prevents models from spending excessive computation for "easy" inputs, where shallow models are sufficient for correct predictions, and therefore reduces the post-training workloads while preserving accuracy.

In this work, we study the robustness of multi-exit language models to *adversarial slowdown*. Recent work [10] showed that, against multi-exit models developed for computer vision tasks, an adversary can craft human-imperceptible input perturbations to offset their computational savings. However, it has not yet been shown that the input-adaptive methods proposed in language domains are susceptible to such input perturbations. It is also unknown why these perturbations cause slowdown and how similar they are to those generated by standard adversarial attacks. Moreover, it is unclear if existing defenses, e.g., adversarial training [43], proposed in the community can mitigate slowdown attacks.

**Our contributions.** To bridge this gap, we *first* develop WAFFLE, a slowdown attack that generates natural adversarial text that bypasses early-exits. We illustrate how our attack works in Figure 1.

37th Conference on Neural Information Processing Systems (NeurIPS 2023).

Based on our finding that existing adversarial text attacks [16, 43] fail to cause significant slowdown, we design a novel objective function that pushes a multi-exit model's predictions at its early-exits toward the uniform distribution. WAFFLE integrates this objective into existing attack algorithms.

*Second*, we systematically evaluate the robustness of three early-exit mechanisms [41, 45, 21] on the GLUE benchmark against adversarial slowdown. We find that WAFFLE significantly offsets the computational savings provided by the mechanisms when each text input is individually subject to perturbations. We also show that methods offering more aggressive computational savings are more vulnerable to our attack.

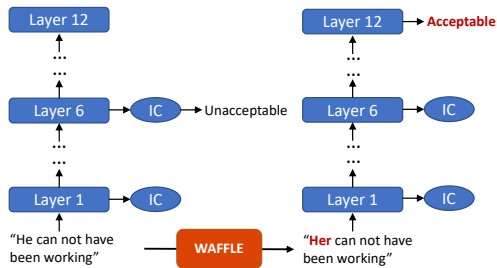

*Third*, we show that WAFFLE can be effective in black-box scenarios. We demonstrate that our attack transfers, i.e., the adversarial texts crafted with limited knowledge about the victim cause slowdown across different models and multi-exit mechanisms. We are also able to find universal slowdown triggers, i.e., input-agnostic sequences of words that reduces the computational savings of multi-exit language models when attached to any text input from a dataset.

Figure 1: **Illustrating adversarial slowdown.** Replacing the word "He" with "Her" makes the resulting text input bypass all 11 ICs (internal classifiers) and leads to misclassification. The text is chosen from the Corpus of Linguistic Acceptability.

*Fourth*, we conduct a linguistic analysis of the adversarial texts WAFFLE crafts. We find that the effectiveness of the attack is not due to the amount of perturbations made on the input text, but rather how perturbations are made. Specifically, we find two critical characteristics present in a vast majority of successful samples: (1) subject-predicate disagreement, meaning that a subject and corresponding verb within a sentence do not match, and (2) the changing of named entities. These characteristics are highlighted in [31], where it was shown that BERT takes both into account when making predictions.

*Fifth*, we test the effectiveness of potential countermeasures against adversarial slowdown. We find that adversarial training [12, 43] is ineffective against WAFFLE. The defended multi-exit models lose efficacy, or they lose significant amounts of accuracy in exchange for aggressive computational savings. In contrast, we show that input sanitization can be an effective countermeasure. This result suggests that future work is needed to develop robust yet effective multi-exit language models.

## 2   Related Work

**Adversarial text attacks on language models.** Szegedy et al. [34] showed that neural network predictions can be *fool*-ed by human-imperceptible input perturbations and called such perturbed inputs *adversarial examples*. While earlier work in this direction studied these adversarial attacks on computer vision models [2, 25], there has been a growing body of work on searching for adversarial examples in language domains as a result of language models gaining more traction. However, adversarial texts are much harder to craft due to the discrete nature of natural language. Attacks on images leverage perturbations derived from computing *gradients*, as they are composed of pixel values forming a near-continuous space, but applying them to texts where each word is in a discrete space is not straightforward. As a result, diverse mechanisms for crafting natural adversarial texts [7, 30, 20, 16, 8, 9, 19] have been proposed. In this work, we show that an adversary can exploit the language model's sensitivity to input text perturbations to achieve a completely different attack objective, i.e., adversarial slowdown. A standard countermeasure against the adversarial input perturbation is *adversarial training* that augments the training data with natural adversarial texts [13, 47, 15, 23, 43]. We also show that adversarial training and its adaptation to our slowdown attacks are ineffective in mitigating the vulnerability to adversarial slowdown.

**Input-adaptive efficient neural network inference.** Neural networks are, while accurate, computationally demanding in their post-training operations. Kaya et al. [17] showed that *overthinking* is one problem—these models use all their internal layers for making predictions on every single input even for the "easy" samples where a few initial layers would be sufficient. Prior work [35, 17, 11] proposed *multi-exit architectures* to mitigate the wasteful effect of overthinking. They introduce multiple internal classifiers (i.e., early exits) to a pre-trained model and fine-tune them on the training data to

make correct predictions. During the inference on an input, these early-exits enable *input-adaptive* inference, i.e., a model stops running forward if the prediction confidence is sufficient at an exit.

Recent work [32, 41, 45, 40, 42, 48, 21, 44] adapted the multi-exit architectures to language models, e.g., BERT [5], to save their computations at inference. DeeBERT [41] and FastBERT [22] have been proposed, both straightforward adaptations of multi-exit architectures to language models. Zhou et al. [45] proposed patience-based early exits (PABEE) and showed that one could achieve efficiency, accuracy, and robustness against natural adversarial texts. Liao et al. [21] presented PastFuture that makes predictions from a global perspective, considering past and future predictions from all the exits. However, no prior work studied the robustness of the computational savings that these mechanisms offer to adversarial slowdown [10]. We design a slowdown attack to *audit* their robustness. More comparisons of our work to the work done in computer vision domains are in Appendix A.

# 3 Our Auditing Method: WAFFLE Attack

## 3.1 Threat Model

We consider an adversary who aims to reduce the computational savings provided by a *victim* multi-exit language model. To achieve this goal, the attacker performs perturbations to a natural test-time text input $x \in \mathcal{S}$. The resulting adversarial text $x'$ needs more layers to process for making a prediction. This attack potentially violates the computational guarantees made by real-time systems harnessing multi-exit language models. For example, the attacker can increase the operational costs of the victim model or push the response time of such systems outside the expected range.

Just like language models deployed in the real-world that accept any user input, we assume that the attacker has the ability to query the victim model with perturbed inputs. We focus on the word-level perturbations as they are well studied and efficient to perform with word embeddings [28]. But it is straightforward to extend our attack to character-level or sentence-level attacks by incorporating the slowdown objective we design in Sec 3.2 into their adversarial example-crafting algorithms.

To assess the slowdown vulnerability, we comprehensively consider both *white-box* and *black-box* settings. The white-box adversary knows all the details of the victim model, such as the training data and the model parameters, while the black-box attacker has limited knowledge of the victim model.

## 3.2 The Slowdown Objective

Most adversarial text-crafting algorithms iteratively apply perturbations to a text input until the resulting text $x'$ achieves a pre-defined goal. The goal here for the standard adversarial attacks is the untargeted misclassification of $x'$, i.e., $f_\theta(x') \neq y$. Existing adversarial text attacks design an objective (or a score) function that quantifies how the perturbation of a word (e.g., substitution or removal) helps the perturbed sample achieve the goal. At each iteration $t$, the attacker considers all feasible perturbations and chooses one that minimizes the objective the most.

We design our score function to quantify how close a perturbed sample $x'$ is to causing the worst-case slowdown. The worst-case we consider here is that $x'$ bypasses all the early exits, and the victim model classifies $x'$ at the final layer. We formulate our score function $s(x', f_\theta)$ as follows:

$$s(x', f_\theta) = \sum_{0 < i \leq K} \left( 1 - \frac{1}{N-1} \mathcal{L}\big(F_i(x'), \hat{y}\big) \right)$$

Here, the score function takes $x'$ as the perturbed text and $f_\theta$ as the victim model. It computes the loss $\mathcal{L}$ between the softmax output of an $i$-th internal classifier $F_i$ and the uniform probability distribution $\hat{y}$ over classes. We use $\ell_1$ loss. $K$ is the number of early-exits, and $N$ is the number of classes.

The score function $s$ returns a value in [0, 1]. It becomes one if all $F_i$ is close to the uniform distribution $\hat{y}$; otherwise, it will be zero. Unlike conventional adversarial attacks, our score function over iterations pushes all $F_i$ to $\hat{y}$ (*i.e.*, the lowest confidence case). Most early-exit mechanisms stop forward pass if $F_i$'s prediction confidence is higher than a pre-defined threshold $T$; thus, $x'$ that achieves the lowest confidence bypasses all the exit points.

## 3.3 The WAFFLE Attack

We finally implement WAFFLE by incorporating the slowdown objective we design into existing adversarial text attacks. In this work, we adapt two existing attacks: TextFooler [16] and A2T [43]. TextFooler is by far the strongest black-box attack [38], and A2T is a gradient-based white-box attack. In particular, A2T can craft natural adversarial texts much faster than black-box attacks; thus, it facilitates adversarial training of language models. We discuss the effectiveness of this in Sec 7.

We describe how we adapt TextFooler for auditing the slowdown risk in Algorithm 1 (see Appendix for our adaptation of A2T). We highlighted our adaptation to the original Textfooler in blue.

---

**Algorithm 1** WAFFLE (based on TextFooler)

---

**Input:** a text input $x = \{w_1, w_2, ..., w_n\}$, its label $y$, the victim model $f_\theta$, its early exits $F_i$, sentence similarity function $Sim(\cdot)$, its threshold $\epsilon$, word embeddings $E$ over the vocabulary $V$, and the attack success threshold $\alpha$.
**Output:** a natural adversarial text $x'$

1:   $x' \leftarrow x$
2:   **for** each word $w_i$ in $x$ **do**
3:      Compute the importance $I_{w_i}$
4:   **end for**
5:   Compose a set $W$ of all words $w_i \in x$ sorted by the descending order of their importance
6:   Remove the stop words from the set $W$
7:   **for** each word $w_i$ in $W$ **do**
8:      Initiate the set of substitute candidates $C$ by computing the top $N$ synonyms; we compute the cosine similarity between $E_{w_i}$ and $E_{w'}$, where $w' \in V$
9:      $C \leftarrow$ POSFiler$(C)$
10:     $C_{final} \leftarrow \{\}$
11:     **for** $c_k$ in $C$ **do**
12:        $x^{temp} \leftarrow$ Replace $w_j$ with $c_k$ in $x'$
13:        **if** $Sim(x^{temp}, x') > \epsilon$ **then**
14:          Add $c_k$ to $C_{final}$
15:          $s_k \leftarrow f_\theta(x^{temp})$
16:        **end if**
17:     **end for**
18:     **if** $\exists c_k$ whose score is $s_k \geq \alpha$ **then**
19:        Keep the candidates $c_k \in C_{final}$
20:        $c^* \leftarrow \text{argmax}_{c \in C_{final}} Sim(x, x^{temp}_{w_j \to c})$
21:        $x' \leftarrow$ Replace $w_j$ with $c^*$ in $x'$
22:        **return** $x'$
23:     **else if** $s_k(x') > min\ s_k$ **then**
24:        $c^* \leftarrow \text{argmax}_{c_k \in C_{final}} s_k$
25:        $x' \leftarrow$ Replace $w_j$ with $c^*$ in $x'$
26:     **end if**
27:   **end for**
28:   **return** $x'$

---

**(line 1–6) Compute word importance.** We first compute the importance of each word $w_i$ in a text input $x$. TextFooler removes each word from $x$ and computes their influence on the final prediction result. It then ranks the words based on their influence. In contrast, we rank the words based on the *increase* in the slowdown objective after each removal. By perturbing only a few words, we can minimize the alterations to $x$ and keep the semantic similarity between $x'$ and $x$. Following the original study, we filter out stop words, *e.g.*, 'the' or 'when', to minimize sentence structure destruction.

**(line 7–9) Choose the set of candidate words for substitution.** The attack then works by replacing a set of words in $x$ with the candidates carefully chosen from $V$. For each word $w_i \in x$, the attack collects the set of $C$ candidates by computing the top $N$ synonyms from $V$ (line 8). It computes the cosine similarity between the embeddings of the word $w_i$ and the word $w' \in V$. We use the same embeddings [28] that the original study uses. TextFooler only keeps the candidate words with the same part-of-speech (POS) as $w_i$ to minimize grammar destruction (line 9).

**(line 10–28) Craft a natural slowdown text.** We then iterate over the remaining candidates and substitute $w_i$ with $c_k$. If the text after this substitution $x^{temp}$ is sufficiently similar to the text before it, $x'$, we store the candidate $c_k$ into $C_{final}$ and compute the slowdown score $s_k$. In the end, we have a perturbed text input $x'$ that is similar to the original input within the $\epsilon$ similarity and the slowdown score $s_k$ (line 10–17). To compute the semantic similarity, we use Universal Sentence Encoder that the original study uses [3].

In line 20–26, if there exists any candidate $c_k$ that already increases the slowdown score $s_k$ over the threshold $\alpha$ we choose the word with the highest semantic similarity among these winning candidates. However, when there is no such candidate, we pick the candidate with the highest slowdown score, substitute the candidate with $w_i$, and repeat the same procedure with the next word $w_{i+1}$. At the end (line 28), TextFooler does not return any adversarial example if it fails to flip the prediction. However, as our goal is causing slowdown, we use this adversarial text even when the score is $s_k \leq \alpha$.

# 4 Auditing the Robustness to Adversarial Slowdown

We now utilize our WAFFLE attack as a vehicle to evaluate the robustness of the computational savings provided by multi-exit language models. Our adaptations of two adversarial text attacks, TextFooler and A2T, represent the black-box and white-box settings, respectively.

**Tasks.** We evaluate the multi-exit language models trained on seven classification tasks chosen from the GLUE benchmark [36]: RTE, MRPC, MNLI, QNLI, QQP, SST-2, and CoLA.

**Multi-exit mechanisms.** We consider three early-exit mechanisms recently proposed for language models: DeeBERT [41], PABEE [45], and Past-Future [21]. In DeeBERT, we take the pre-trained BERT and fine-tune it on the GLUE tasks. We use the pre-trained ALBERT [18] for PABEE and Past-Future. To implement these mechanisms, we use the source code from the original studies. We describe all the implementation details, e.g., the hyper-parameter choices, in Appendix.

**Metrics.** We employ two metrics: *classification accuracy* and *efficacy* proposed by Hong et al. [10]. We compute both the metrics on the test-set $S$ or the adversarial texts crafted on $S$. Efficacy is a standardized metric that quantifies a model's ability to use its early exits. It is close to one when a higher percentage of inputs exit at an early layer; otherwise, it is 0. To quantify the robustness, we report the changes in accuracy and efficacy of a clean test-set $S$ and $S$ perturbed using WAFFLE.

## 4.1 Multi-exit Language Models Are Not Robust to Adversarial Slowdown

| Attack | Metric | GLUE Task | | | | | | |
|---|---|---|---|---|---|---|---|---|
| | | RTE | MNLI | MRPC | QNLI | QQP | SST-2 | CoLA |
| **DeeBERT (BERT-base)** | | | | | | | | |
| TF | Acc. | $63 \to 48$ | - | $82 \to 75$ | $88 \to 78$ | $92 \to 67$ | - | $79 \to 57$ |
| | Eff. | $0.34 \to 0.32$ | - | $0.35 \to 0.32$ | $0.35 \to 0.33$ | $0.36 \to 0.40$ | - | $0.34 \to 0.20$ |
| A2T | Acc. | $63 \to 52$ | - | $82 \to 75$ | $88 \to 81$ | $92 \to 74$ | - | $79 \to 66$ |
| | Eff. | $0.34 \to 0.32$ | - | $0.35 \to 0.33$ | $0.35 \to 0.35$ | $0.36 \to 0.41$ | - | $0.34 \to 0.29$ |
| WAFFLE (TF) | Acc. | $63 \to 51$ | - | $82 \to 61$ | $88 \to 62$ | $92 \to 69$ | - | $79 \to 70$ |
| | Eff. | $0.34 \to 0.11$ | - | $0.35 \to 0.09$ | $0.35 \to 0.10$ | $0.36 \to 0.22$ | - | $0.34 \to 0.13$ |
| WAFFLE (A2T) | Acc. | $63 \to 57$ | - | $82 \to 75$ | $88 \to 78$ | $92 \to 83$ | - | $79 \to 73$ |
| | Eff. | $0.34 \to 0.19$ | - | $0.35 \to 0.17$ | $0.35 \to 0.19$ | $0.36 \to 0.30$ | - | $0.34 \to 0.24$ |
| **PABEE (ALBERT-base)** | | | | | | | | |
| TF | Acc. | $79 \to 34$ | $83 \to 25$ | $87 \to 37$ | $91 \to 33$ | $92 \to 31$ | $93 \to 22$ | $82 \to 5$ |
| | Eff. | $0.24 \to 0.22$ | $0.28 \to 0.17$ | $0.32 \to 0.21$ | $0.31 \to 0.18$ | $0.37 \to 0.27$ | $0.37 \to 0.26$ | $0.32 \to 0.23$ |
| A2T | Acc. | $79 \to 57$ | $83 \to 52$ | $87 \to 63$ | $91 \to 71$ | $92 \to 61$ | $93 \to 76$ | $82 \to 38$ |
| | Eff. | $0.24 \to 0.22$ | $0.28 \to 0.21$ | $0.32 \to 0.26$ | $0.31 \to 0.27$ | $0.37 \to 0.31$ | $0.37 \to 0.32$ | $0.32 \to 0.23$ |
| WAFFLE (TF) | Acc. | $79 \to 57$ | $83 \to 38$ | $87 \to 47$ | $91 \to 51$ | $92 \to 67$ | $93 \to 50$ | $82 \to 48$ |
| | Eff. | $0.24 \to 0.09$ | $0.28 \to 0.05$ | $0.32 \to 0.08$ | $0.31 \to 0.06$ | $0.37 \to 0.17$ | $0.37 \to 0.08$ | $0.32 \to 0.08$ |
| WAFFLE (A2T) | Acc. | $79 \to 72$ | $83 \to 69$ | $87 \to 73$ | $91 \to 82$ | $92 \to 79$ | $93 \to 85$ | $82 \to 60$ |
| | Eff. | $0.24 \to 0.17$ | $0.28 \to 0.18$ | $0.32 \to 0.21$ | $0.32 \to 0.23$ | $0.37 \to 0.27$ | $0.37 \to 0.29$ | $0.32 \to 0.19$ |
| **PastFuture (ALBERT-base)** | | | | | | | | |
| TF | Acc. | $74 \to 41$ | $86 \to 42$ | $88 \to 36$ | $92 \to 52$ | $92 \to 50$ | - | - |
| | Eff. | $0.52 \to 0.46$ | $0.50 \to 0.24$ | $0.50 \to 0.24$ | $0.50 \to 0.19$ | $0.52 \to 0.35$ | - | - |
| A2T | Acc. | $74 \to 58$ | $86 \to 59$ | $88 \to 58$ | $92 \to 74$ | $92 \to 64$ | - | - |
| | Eff. | $0.52 \to 0.49$ | $0.50 \to 0.32$ | $0.50 \to 0.31$ | $0.50 \to 0.35$ | $0.52 \to 0.43$ | - | - |
| WAFFLE (TF) | Acc. | $74 \to 51$ | $86 \to 45$ | $88 \to 42$ | $92 \to 58$ | $92 \to 64$ | - | - |
| | Eff. | $0.51 \to 0.17$ | $0.50 \to 0.05$ | $0.50 \to 0.15$ | $0.50 \to 0.07$ | $0.52 \to 0.25$ | - | - |
| WAFFLE (A2T) | Acc. | $74 \to 64$ | $86 \to 67$ | $88 \to 72$ | $92 \to 83$ | $92 \to 79$ | - | - |
| | Eff. | $0.52 \to 0.36$ | $0.50 \to 0.26$ | $0.50 \to 0.29$ | $0.50 \to 0.33$ | $0.52 \to 0.39$ | - | - |

Table 1: **Robustness of multi-exit language models to our slowdown attacks.** WAFFLE significantly reduces the computational savings offered by DeeBERT, PABEE, and PastFuture. In each cell, we report the accuracy (acc.) and efficacy (eff.) on a clean test set and its corresponding adversarial texts generated by the four attacks ($\to$ denotes going from the clean test set to the adversarial texts).

Table 1 shows our evaluation results. Following the original studies, we set the early-exit threshold, i.e., entropy or patience, so that multi-exit language models have 0.33–0.5 efficacy on the clean test set (see Appendix for more details). We use four adversarial attacks: two standard adversarial attacks, TextFooler (TF) and A2T, and their adaptations: WAFFLE (TF) and WAFFLE (A2T). We perform

these attacks on the entire test-set and report the changes in accuracy and efficacy. In each cell, we include their flat changes and the values computed on the clean and adversarial data in parenthesis.

**Standard adversarial attacks are ineffective in auditing slowdown.** We observe that the standard attacks (TF and A2T) are ineffective in causing a significant slowdown. In DeeBERT, those attacks cause negligible changes in efficacy (-0.05–0.14), while they inflict a large accuracy drop (7%–25%). Against PABEE and PastFuture, we find that the changes are slightly higher than those observed from DeeBERT (*i.e.*, 0.02–0.13 and 0.03–0.31). We can observe slowdowns in PastFuture, but this is not because the standard attacks are effective in causing slowdowns. This indicates the mechanism is more sensitive to input changes, which may lead to greater vulnerability to adversarial slowdown.

**WAFFLE is an effective auditing tool for assessing the slowdown risk.** We show that our slowdown attack can inflict significant changes in efficacy. In DeeBERT and PABEE, the attacks reduce the efficacy 0.06–0.26 and 0.07–0.29, respectively. In PastFuture, we observe more reduction in efficacy 0.13–0.45. These multi-exit language models are designed to achieve an efficacy of 0.33–0.5; thus its reduction up to 0.29–0.45 means a complete offset of their computational savings.

**The more complex a mechanism is, the more vulnerable it is to adversarial slowdown.** WAFFLE causes the most significant slowdown on PastFuture, followed by PABEE and DeeBERT. PastFuture stops forwarding based on the predictions from past exits and the estimated predictions from future exits. PABEE also uses patience, i.e., how often we observe the same decision over early-exits. They enable more *aggressive* efficacy compared to DeeBERT, which only uses entropy. However, this aggressiveness can be exploited by our attacks, e.g., introducing inconsistencies over exit points; thus, PABEE needs more layers to make a prediction.

### 4.2 Sensitivity to Attack Hyperparameter

The key hyperparameter of our attack, the attack success threshold ($\alpha$), determines the magnitude of the scores pursued by WAFFLE while crafting adversarials. The score measures how uniform all output distributions of $F_i$ are. A higher $\alpha$ pushes WAFFLE to have a higher slowdown score before returning a perturbed sample. Figure 2 shows the accuracy and efficacy of all three mechanisms on QNLI against $\alpha$ in [0.1, 1]. We show that as $\alpha$ increases, the slowdown (represented as a decrease in efficacy) increases, and the accuracy decreases.

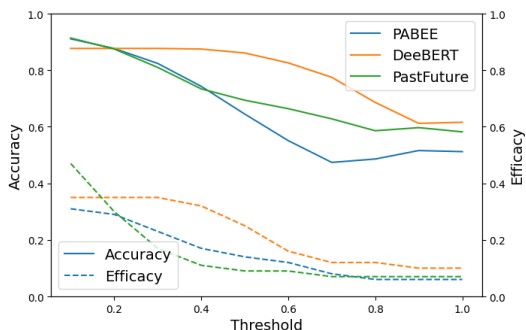

Figure 2: **The impact of $\alpha$ on accuracy and efficacy.** Taking each model's results on QNLI, as $\alpha$ is increased, the accuracy and efficacy decrease.

In addition, as $\alpha$ increases, the rate of decrease in accuracy and efficacy decreases. Note that in PastFuture, when $\alpha \geq 0.4$ the rate at which efficacy decreases drops by a large margin. The same applies to accuracy, and when $\alpha \geq 0.8$, accuracy surprisingly increases, a potentially undesirable outcome. Moreover, when $\alpha \geq 0.8$ efficacy does not decrease any further, which potentially wastes computational resources as the time required to craft samples increases greatly as $\alpha$ is increased.

## 5   Practical Exploitation of WAFFLE in Black-box Settings

In Sec 4, we show in the worst-case scenarios, multi-exit language models are not robust to adversarial slowdown. We now turn our attention to black-box settings where an adversary does not have full knowledge of the victim's system. We consider two attack scenarios: (1) *Transfer-based attacks* where an adversary who has the knowledge of the training data trains *surrogate* models to craft adversarial texts and use them to attack the victim models. (2) *Universal attacks* where an attacker finds a set of *trigger* words that can inflict slowdown when attached to any test-time inputs. We run these experiments across various GLUE tasks and show the results from the RTE and QQP datasets. We include all our results on different datasets and victim-surrogate combinations in the Appendix.

**Transferability of WAFFLE.** We first test if our attack is transferable in three different scenarios: (1) Cross-seed; (2) Cross-architecture; and (3) Cross-mechanism. Table 2 summarizes our results.

| Model | Arch. | Mechanism | Scenario | Type | RTE | | QQP | |
|---|---|---|---|---|---|---|---|---|
| | | | | | Acc. | Eff. | Acc. | Eff. |
| S | BERT | PastFuture | Cross-seed | S→S | 66 → 52 | 0.47 → 0.11 | 91 → 72 | 0.50 → 0.26 |
| V | BERT | PastFuture | | S→V | 66 → 55 | 0.50 → 0.25 | 91 → 72 | 0.51 → 0.33 |
| S | BERT | PABEE | Cross-arch. | S→S | 66 → 49 | 0.22 → 0.08 | 91 → 69 | 0.35 → 0.16 |
| V | ALBERT | PABEE | | S→V | 77 → 55 | 0.22 → 0.21 | 91 → 74 | 0.36 → 0.34 |
| S | BERT | PABEE | Cross-mech. | S→S | 66 → 49 | 0.22 → 0.08 | 91 → 69 | 0.35 → 0.16 |
| V | BERT | PastFuture | | S→V | 66 → 55 | 0.50 → 0.29 | 91 → 74 | 0.51 → 0.38 |

S = Surrogate model; V = Victim model

Table 2: **Transfer-based attack results.** Results from the cross-seed, cross-mechanism, and cross-architecture experiments on RTE and QQP. In all experiments, we craft adversarial texts on the surrogate model (S) and then evaluated on both the surrogate (S→S) and victim (S→V) models.

*Cross-seed.* Both the model architecture and early-exit mechanism are identical for the surrogate and the victim models. In RTE and QQP, our transfer attack (S→V) demonstrates a significant slowdown on the victim model, resulting in a reduction in efficacy of 0.25 and 0.18, respectively. In comparison to the white-box scenarios (S→S), these attacks achieve approximately 50% effectiveness.

*Cross-architecture.* We vary the model architecture, using either BERT or ALBERT, while keeping the early-exit mechanism (PABEE) the same. Across the board, we achieve the lowest transferability among the three attacking scenarios, with a reduction in efficacy of 0.01 in RTE and 0.02 in QQP, respectively. This indicates that when conducting transfer-based attacks, the matching of the victim and surrogate models' architectures has a greater impact than the early-exit mechanism.

*Cross-mechanism.* We now vary the early-exit mechanism used by the victim and surrogate models while the architecture (BERT) remains consistent. In QQP and RTE, we cause significant slowdown to the victim model (a reduction in efficacy of 0.21 and 0.13, respectively), even when considering the relational speed-up offered by different mechanisms (e.g., PastFuture offers more computational savings than PABEE and DeeBERT). The slowdown is comparable to the white-box cases (S→S).

**Universal slowdown triggers.** If the attacker is unable to train surrogate models, they can find a few words (i.e., a trigger) that causes slowdown to any test-time inputs when attached. Searching for such a trigger does not require the knowledge of the training data. To demonstrate the practicality of this attack, we select 1000 random words from BERT's vocabulary and compute the total slowdown across 10% of the SST-2 test dataset by appending each vocab word to the beginning of every sentence. We then choose the word that induces the highest slowdown and evaluate it against the entire test dataset. We find that the most effective word, "unable", reduces efficacy by 9% and accuracy by 14% when appended to the beginning of all sentences once. When appended three times successively (i.e. "unable unable unable . . . "), the trigger reduces efficacy by 18% and accuracy by 3%.

# 6 Lingusitic Analysis of Our Adversarial Texts

To qualitatively analyze the text generated by WAFFLE, we first consider how the number of perturbations applied to an input text affects the slowdown it induces. We choose samples crafted against PastFuture [21], due to it being the most vulnerable to our attack. We select the datasets that induce the most and least slowdown, MNLI and QQP, respectively, in order to conduct a well-rounded analysis. Using 100 samples randomly drawn from both datasets, we record the percentage of words perturbed by WAFFLE and the consequent increase in exit layer. In Figure 3, we find that there is no relationship between the percentage of words perturbed and the increase in exit layer. It is not the number of perturbations made that affects slowdown, but rather how perturbations are made.

In an effort to find another explanation for why samples crafted by WAFFLE induce slowdown on multi-exit models, we look to analyze the inner workings of BERT. Through the qualitative analysis performed by Rogers et al. [31], we find particular interest in two characteristics deemed of high importance to BERT when it makes predictions: (1) subject-predicate agreement and (2) the changing of named entities. In our experiment, these characteristics are incredibly prevalent in successful attacks. Particularly, we find that the score assigned by WAFFLE is much higher when the subject and predicate of a sentence do not match, or a named entity is changed. In addition, WAFFLE

| Task | Original Text | Perturbed Text | Change |
|------|--------------|----------------|--------|
| **RTE** | Sentence1: Mount Olympus towers up from the center of the earth. Sentence2: Mount Olympus is in the center of the earth. | Sentence1: **Install** Olympus towers up from the **facilities** of the **planet**. Sentence2: Mount Olympus is in the center of the earth. | 8→12 |
| **MNLI** | Premise: I'll twist him, sir. Hypothesis: I'll make him straight | Premise: I'll **bending** him, sir. Hypothesis: I'll **implement** him **consecutive**. | 8→12 |
| **MRPC** | Sentence1: Ms Stewart, the chief executive, was not expected to attend. Sentence2: Ms Stewart, 61, ..., did not attend. | Sentence1: **Lena** Stewart, the chief **execute**, was not **scheduled** to **help**. Sentence2: Ms Stewart, 61, ..., did not attend. | 7→12 |
| **QNLI** | Question: Where did the Exposition take place? Sentence: This World's Fair devoted a building to electrical exhibits. | Question: **Whereby** did the **Shows takes** place? Sentence: This World's Fair **devoting** a building to electrical exhibits. | 7→12 |
| **QQP** | Question1: Why do we need to philosophize? Question2: Why do we need to philosophize with others? | Question1: Why do we need to philosophize? Question2: Why **got** we **needing** to philosophize with others? | 10→12 |
| **SST-2** | it's a cookie-cutter movie, a cut-and-paste job. | it's a cookie-cutter **cinematography**, a cut-and-paste **worked**. | 7→12 |
| **CoLA** | I'll work on it if I can. | I'll **task** on it if **me** can. | 7→12 |

Table 3: **Adversarial texts generated by WAFFLE.** An example of text perturbed by WAFFLE from each dataset, using samples crafted for PABEE. They show how subject-predicate disagreement and the changing of named entities can push previously early-exited samples to the final output layer.

often changes non-verbs into verbs or removes verbs entirely, causing further subject-predicate discrepancies. Table 3 shows an example of these characteristics across all the GLUE tasks. Note that subject-predicate agreement appears much more often.

To quantify the prevalence of these characteristics, we analyze 100 adversarial texts, from QQP crafted on DeeBert, most effective in inducing slowdowns and count the number of samples containing subject-predicate disagreement or a changed named entity. Of the 100 samples, 84% had a subject-predicate disagreement and 31% a changed named entity. An important note is that we see smaller percentage points in a changed named entity than in a subject-predicate disagreement, as not all samples have a named entity.

We believe that these two characteristics induce a high amount of slowdown because they can push samples toward out-of-distribution. BERT was trained to have an acute understanding of language and its nuances. It is therefore not expected to have the disagreement between the subject and predicate of a sentence in its training data (unless that is the task it is trained for e.g. CoLA). Also, according to Rogers et al. [31], BERT likely lacks a general idea of named entities, providing an explanation as to why the model is less confident in an answer when a named entity is changed.

Moreover, we analyze samples produced by the base attacks (TF and A2T), and compare them with samples produced by WAFFLE. Surprisingly, we find a great deal of similarity, particularly with regard to the usage of the two characteristics detailed above, despite the base attacks inducing negligible slowdown. A potential explanation for this is that both types of attacks construct out-of-distribution samples, but change a model's confidence to varying extents. The base attacks likely increase confidence in the wrong answer to a much larger degree than WAFFLE, encouraging early-exits with mispredictions in a majority of cases. In contrast, WAFFLE works by reducing confidence across all of the classes.

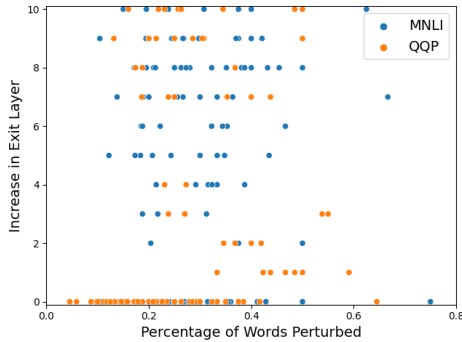

Figure 3: **Visualizing the relationship between words perturbed and adversarial slowdown induced.** Taking 100 random samples from MNLI and QQP (crafted against the PastFuture model), the percentage of words that WAFFLE perturbed and the resulting increase in exit layer is plotted (a greater increase in exit layer indicates greater slowdown). The figure shows that the percentage of words perturbed has *no* influence over a sample's ability to induce slowdown.

## 7   Potential Countermeasures

We now test the effectiveness of potential countermeasures against adversarial slowdown. We first evaluate *adversarial training* (AT), a standard countermeasure that reduces the sensitivity of a model to adversarial input perturbations. We then discuss a way to *sanitize* the perturbations applied to inputs by running off-the-shelf tools, such as a grammar-checking tool, to remove attack artifacts.

To evaluate, we use the AT proposed by Yoo and Qi [43]. This AT requires significantly fewer computational resources; thus, it is better suited for adversarially training large language models. We

run AT with two different natural adversarial texts, those crafted by A2T and by WAFFLE (adapted from A2T). During training, we attack 20% of the total samples in each batch. We run our experiments with PABEE trained on RTE and SST-2. We first set the patience to 6 (consistent with the rest of our experiments), but we set it to 2 for the models trained with WAFFLE (A2T). Once trained, we examine the defended models with attacks stronger than A2T: TextFooler (TF) and WAFFLE (Ours).

**AT is ineffective against our slowdown attacks.** Table 4 shows that AT significantly reduces the efficacy of a model. Compared to the undefended models, the defended models achieve $\sim$0 efficacy. As these models do not utilize early exits, they seem robust to our attacks. But certainly, it is not desirable. It is noticeable that the defended models still suffer from a large accuracy drop. We then decided to set the patience to two, i.e., the multi-exit language models use early-exits more aggressively. The defended models have either very low efficacy or accuracy, and our attacks can reduce both.

| AT | P | Attack | RTE | | SST-2 | |
|---|---|---|---|---|---|---|
| | | | Acc. | Eff. | Acc. | Eff. |
| A2T | 6 | TF | $81 \rightarrow 8$ | $0.04 \rightarrow 0.04$ | $92 \rightarrow 5$ | $0.04 \rightarrow 0.04$ |
| | | Ours | $81 \rightarrow 60$ | $0.04 \rightarrow 0.04$ | $92 \rightarrow 59$ | $0.04 \rightarrow 0.04$ |
| | 2 | TF | $72 \rightarrow 24$ | $0.13 \rightarrow 0.13$ | $89 \rightarrow 10$ | $0.08 \rightarrow 0.07$ |
| | | Ours | $72 \rightarrow 59$ | $0.13 \rightarrow 0.14$ | $89 \rightarrow 56$ | $0.08 \rightarrow 0.07$ |
| A2T (Ours) | 6 | TF | $78 \rightarrow 7$ | $0.04 \rightarrow 0.04$ | $92 \rightarrow 6$ | $0.04 \rightarrow 0.04$ |
| | | Ours | $78 \rightarrow 56$ | $0.04 \rightarrow 0.04$ | $92 \rightarrow 61$ | $0.04 \rightarrow 0.04$ |
| | 2 | TF | $53 \rightarrow 53$ | $0.65 \rightarrow 0.65$ | $90 \rightarrow 7$ | $0.05 \rightarrow 0.04$ |
| | | Ours | $53 \rightarrow 53$ | $0.65 \rightarrow 0.65$ | $90 \rightarrow 57$ | $0.05 \rightarrow 0.04$ |

Table 4: **Effectiveness of AT.** AT is ineffective against WAFFLE. The defended models completely lose the computational efficiency (*i.e.*, they have $\sim$0 efficacy), even with the aggressive setting with the patience of 2. **P** is the patience.

This result highlights a trade-off between being robust against adversarial slowdown and being efficient. We leave further explorations as future work.

**Input sanitization can be a defense against adversarial slowdown.** Our linguistic analysis in Sec 6 shows that the subject-predicate discrepancy is one of the root causes of the slowdown. Building on this insight, we test if sanitizing the perturbed input before feeding it into the models can be a countermeasure against our slowdown attacks. We evaluate this hypothesis with two approaches.

We first use OpenAI's ChatGPT[1], a conversational model where we can ask questions and get answers. We manually query the model with natural adversarial texts generated by WAFFLE (TF) and collect the revised texts. Our query starts with the prompt "Can you fix all of these?" followed by perturbed texts in the subsequent lines. We evaluate with the MNLI and QQP datasets, on 50 perturbed test-set samples randomly chosen from each. We compute the changes in accuracy and average exit number on the perturbed samples and their sanitized versions. We compute them on the PastFuture models trained on the datasets. Surprisingly, we find that the inputs sanitized by ChatGPT greatly recovers both accuracy and efficacy. In MNLI, we recover the accuracy by 12 percentage points (54%→66%) and reduce the average exit number by 4 (11.5→7.5). In QQP, the accuracy is increased by 24 percentage points (56%→80%), and the average exit number is reduced by 2.5 (9.6→7.1).

We also test the effectiveness of additional grammar-checking tools, such as Grammarly[2] and language_tool_python[3], in defeating our slowdown attacks. We run this evaluation using the same settings as mentioned above. We feed the adversarial texts generated by our attack into Grammarly and have it correct them. Note that we only take the Grammarly recommendations for the correctness and disregard any other recommendations, such as for clarity. We find that the inputs sanitized by Grammarly still suffer from a significant accuracy loss and slowdown. In MNLI, both the accuracy and the average exit number stay the same 56%→58% and 9.6→9.5, respectively. In QQP, we observe that there is almost no change in accuracy (54%→58%) or the average exit number (11.5→11.5).

# 8 Conclusion

This work shows that the computational savings that input-adaptive multi-exit language models offer are *not* robust against adversarial slowdown. To evaluate, we propose WAFFLE, an adversarial text-crafting algorithm with the objective of bypassing early-exit points. WAFFLE significantly reduces the computational savings offered by those models. More sophisticated input-adaptive mechanisms suited for higher efficacy become more vulnerable to slowdown attacks. Our linguistic analysis

---

[1]ChatGPT: https://openai.com/blog/chatgpt/

[2]Grammarly: https://www.grammarly.com/

[3]Language Tool (Python): https://github.com/jxmorris12/language_tool_python

exposes that it is not about the magnitude of perturbations but because pushing an input outside the distribution on which a model is trained is easy. We also show the limits of adversarial training in defeating our attacks and the effectiveness of input sanitization as a defense. Our results suggest that future research is required to develop efficient yet robust input-adaptive multi-exit inference.

## 9   Limitations, Societal Impacts, and Future Work

As shown in our work, word-level perturbations carefully calibrated by WAFFLE make the resulting natural adversarial texts offset the computational savings multi-exit language models provide. However, there have been other types of text perturbations, e.g., character-level [6, 1] or sentence-level perturbations [37]. We have not tested whether an adversary can adapt them to cause slowdowns. If these new attacks are successful, we can hypothesize that some other attributes of language models contribute to lowering the confidence of the predictions made by internal classifiers (early-exit points). It may also render potential countermeasures, such as input sanitization, ineffective. Future work is needed to investigate attacks exploiting different perturbations to cause adversarial slowdown.

To foster future research, we developed WAFFLE in an open-source adversarial attack framework, TextAttack [27]. This will make our attacks more accessible to the community. A concern is that a potential adversary can use those attacks to push the behaviors of systems that harness multi-exit mechanisms outside the expectations. But we believe that our offensive measures will be adopted broadly by practitioners and have them audit such systems before they are publicly available.

We have also shown that using state-of-the-art conversational models, such as ChatGPT, to sanitize perturbed inputs can be an effective defense against adversarial slowdown. But it is unclear what attributes of those models were able to remove the artifacts (i.e., perturbations) our attack induces. Moreover, the fact that this defense heavily relies on the referencing model's capability that the victim cannot control may give room for an adversary to develop stronger attacks in the future.

It is also possible that when using conversational models online as a potential countermeasure, there will be risks of data leakage. However, our proposal does not mean to use ChatGPT as-is. Instead, since other input sanitation (Grammarly) failed, we used it as an accessible tool for input sanitization via a conversational model as a proof-of-concept that it may have effectiveness as a defense. Alternatively, a defender can compose input sanitization as a completely in-house solution by leveraging off-the-shelf models like Vicuna-13B [4]. We leave this exploration as future work.

An interesting question is to what extent models like ChatGPT offer robustness to the *conventional* adversarial attacks that aim to reduce a model's utility in the inference time. But this is not the scope of our work. While the conversational models we use offer some robustness to our slowdown attacks with fewer side-effects, it does not mean that this direction is bulletproof against all adversarial attacks and/or adaptive adversaries in the future. Recent work shows two opposite views about the robustness of conversational models [39, 49]. We envision more future work on this topic.

We find that the runtime of the potential countermeasures we explore in Sec 7 is higher than the average inference time of *undefended* multi-exit language models. This would make them useless from a pure runtime standpoint. However, we reason that the purpose of using these defenses was primarily exploratory, aiming to understand further why specific text causes more slowdown and how modifying such text can revert this slowdown. Moreover, input sanitization is already used in commercial models. Claude-2[4], a conversational model similar to ChatGPT, already employs input-filtering techniques, which we believe, when combined together, is a promising future work direction. Defenses with less computational overheads must be an important area for future work.

Overall, this work raises an open-question to the community about the feasibility of *input-adaptive* efficient inference on large language models. We believe future work is necessary to evaluate this feasibility and develop a mechanism that kills two birds (efficacy and robustness) with one method.

---

[4]https://claude.ai

## Acknowledgements

We thank the anonymous reviewers for their constructive feedback. Zachary Coalson and Sanghyun Hong are partially supported by the Google Faculty Research Award and the Samsung Global Research Outreach (GRO) program. Gabriel Ritter and Rakesh Bobba are partially supported by the U.S. Department of Transportation. The findings and conclusions in this work are those of the author(s) and do not necessarily represent the views of the funding agency.

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

# A    Comparison to Related Attacks in Prior Work

Here, we expand upon our discussion in Sec 2 and discuss the novelty of our slowdown attack compared to the attacks developed in prior work [1, 33, 10].

Prior work [1, 33] has shown that an adversary can increase the energy consumption of language models in modern computing hardware via "sponge" examples. These inputs exploit computational properties of hardware or tokenization, e.g., input dimensionality and/or activation sparsity, to increase the inference runtime. In contrast, our attack is hardware-agnostic and targets multi-exit language models, a new algorithm for efficient language model computations. No prior work has been done on adversarial slowdowns in the context of multi-exit language models, and our attack is the first that generates natural adversarial text that bypasses the early-exit layers of multi-exit models.

Compared to the slowdown attacks in the computer vision domain [10], we also highlight the unique challenges we address: (1) Against language models, we often do not have access to input gradients (which is straightforward in attacks against computer vision models). We thus need to design a new slowdown objective compatible with non-gradient-based attacks. (2) We must bound the values of our slowdown objective within [0, 1]. The objective used in the prior work [10] is unbounded to [0, $\infty$]; thus, a straightforward adaptation of this objective for adversarial text-attack algorithms leads to unbounded perturbations, and the resulting text completely differs from the original one. (3) The attack against language models works with discrete text inputs; not all embedding-level perturbations we compute exist as words, and small changes to input (word, characters) can result in large logit changes. We must search for candidate words (or word combinations) for substitution.

# B    Experimental Setup in Detail

Here, we describe our experimental setup in detail. We implement all the multi-exit mechanisms and our attacks using Python v3.9[5] and PyTorch v1.10[6] that supports CUDA 11.7 for accelerating computations by using GPUs. We take the pre-trained language models (*i.e.*, BERT and ALBERT) from Hugging Face[7] and fine-tune them on GLUE benchmarks. Our experiments run on a machine equipped with Intel Xeon Processor with 48 cores, 64GB memory and 8 Nvidia A40 GPUs.

**Multi-exit language models.** All the early-exit mechanisms we employ, *i.e.*, DeeBERT, PABEE, and PastFuture, takes a pre-trained language model, attaches an internal classifier (*i.e.*, an early-exit) to each internal layer, and fine-tune the entire model on a task. We choose the pre-trained BERT ('bert-base-uncased') and ALBERT ('albert-base-v2') from Hugging Face. We fine-tune them on seven different GLUE tasks for five epochs. We choose a batch-size from 32, 64, 128 and a learning rate from 1e-5, 2e-5, 3e-5, 4e-5, 5e-5. We perform hyper-parameter sweeping over all the combinations and select the models that provide the best accuracy for each task. We select the early-exit thresholds based on the values in the original studies. In DeeBERT [41], we pick the entropy that offers $1.5\times$ computational speedup. In PABEE [45], we choose the patience value of 6. In PastFuture [21], we set the entropy values where we achieve $2\times$ speedup.

**Choice of the slowdown metric.** Prior work on early-exit mechanisms uses two metrics: *wall-clock time* and *speedup*. DeeBERT uses wall-clock time, but it is not a desirable metric as the metric depends on the choice of hardware or software libraries, such as deep learning frameworks. PABEE and PastFuture propose speedup, a ratio between the total number of layers and the number of layers required to make a prediction. They compute this ratio over the entire test-set samples and report the average value. However, it is also not an accurate estimation of the computational savings, as depending on model architectures, the number of parameters in a layer and the way it computes the inputs could be different. As a result, we employ *efficacy* that counts the number of floating-point computations. Note that BERT and ALBERT are both stacks of Transformer layers; this, luckily, the speedup is the inverse of the efficacy.

# C    The WAFFLE Attack Based on A2T

---

[5]Python: https://www.python.org
[6]PyTorch: https://pytorch.org
[7]Hugging Face: https://huggingface.co

We show how we adapt A2T [43] for auditing the slowdown risk in Algorithm 2. We highlighted our adaptation to A2T in blue.

**(line 1–2) Compute word importance.** We first compute the importance of each word $w_i$ in a text input $x$. The procedure is the same as shown in Sec 3.3; we remove each word from $x$ and compute the influence on the slowdown objective. We then rank the words based on how much each removal increases the slowdown score $s_i$. We also filter out stop words, *e.g.*, 'the' or 'when'.

**(line 3–13) Craft a natural adversarial text.** The attack then works by replacing a set of words in $x$ with the candidates carefully chosen by $T(x^*, i)$. The transformation function $T$ selects the top 20 synonyms that has the similar embeddings [28], based on the cosine similarity. We only keep the candidates with the same part-of-speech as $w_i$ to minimize grammar destruction.

---

**Algorithm 2** WAFFLE (based on A2T)

---
**Input:** a text input $x = \{w_1, w_2, ..., w_n\}$; the victim model $f_\theta$; a transformation module $T(x, i)$ that perturbs $x$ by replacing $w_i$; and the success threshold $\alpha$.
**Output:** a natural adversarial text $x^*$
1: Calculate $I(w_i)$ for all words $w_i$ by making one forward and backward pass.
2: $R \leftarrow$ ranking $r_1, r_2, ..., r_n$ of words $w_1, ..., w_n$ by descending importance
3: $x^* \leftarrow x$
4: **for** $i = r_1, r_2, ..., r_n$ in $R$ **do**
5:     $X_{cand} \leftarrow T(x^*, i)$
6:     **if** $X_{cand} \neq \emptyset$ **then**
7:         $x^* \leftarrow \operatorname{argmax}_{x^t \in X_{cand}} s_i(x^t, f_\theta)$
8:         **if** $s_i \geq \alpha$ **then**
9:             **return** $x^*$
10:         **end if**
11:     **end if**
12: **end for**
13: **return** $x^*$

---

We then substitute $w_i$ with the candidate that maximizes the slowdown score $s_i(x^t, f_\theta)$ after the substitution. If the text after this substitution $x^*$ increases the slowdown score over the threshold $\alpha$, we return $x^*$. However, when there is no such candidate, we pick the candidate with the highest slowdown score, substitute the candidate with $w_i$, and repeat the same procedure with the next word $w_{i+1}$. In the end, we return $x^*$ even when the slowdown score does not meet the threshold $\alpha$.

# D   Data and Code Availability

As a part of the reproducible research practice, we release our data and source code along with our submission. Our WAFFLE attacks are implemented using TextAttack [27], a Python framework for testing a model's robustness to adversarial attacks. We also include our attacks on the TextAttack repo[8]. This will encourage practitioners and AI-system engineers developing (or employing) input-adaptive efficient inference mechanisms to test their robustness to adversarial slowdown.

# E   More Results on Transferability of WAFFLE

Here we provide further results from our transferability experiments in Sec 5. For the cross-seed and cross-mechanism attacks, we show all victim-surrogate combinations involving all three early-exit mechanisms. For the cross-architecture attack, we show our results on all seven GLUE tasks.

Table 5 shows the entire results from the cross-seed transfer-based attacks. Examining all the three early-exit mechanisms, attacking the victim model using the adversarial texts crafted on the surrogate models causes approximately 50% of the slowdown induced when attacking the surrogate directly.

Table 6 shows all results from the cross-mechanism attack scenario. In a majority of victim-surrogate combinations, we observe the slowdown similar to the cross-seed scenario. This makes sense, as only the early-exit mechanisms differ which account for a small number of parameters relative to the entire model. The result also imply that even if an attacker does not know the specific early-exit mechanism of the target model, high slowdown can still be induced.

Table 7 shows all results from the cross-architecture attack. We run our experiments with the entire GLUE tasks. Compared to the previous two attacks, the cross-architecture attack is less effective. This implies that knowing the target's architecture is a critical when exploiting adversarial transferability. If architecture is known, a strong attack can still be launched even if the early-exit mechanism and parameter values remain unknown to the attacker.

---

[8]TextAttack: https://github.com/QData/TextAttack

| Model | Arch. | Mechanism | Type | RTE | | QQP | |
|---|---|---|---|---|---|---|---|
| | | | | Acc. | Eff. | Acc. | Eff. |
| S | BERT | DeeBERT | S → S | 67 → 55 | 0.32 → 0.11 | 91 → 76 | 0.32 → 0.18 |
| V | BERT | DeeBERT | S → V | 64 → 52 | 0.36 → 0.23 | 91 → 77 | 0.35 → 0.30 |
| S | BERT | PABEE | S → S | 66 → 49 | 0.22 → 0.08 | 91 → 69 | 0.35 → 0.16 |
| V | BERT | PABEE | S → V | 65 → 61 | 0.22 → 0.14 | 91 → 71 | 0.35 → 0.26 |
| S | BERT | PastFuture | S → S | 66 → 52 | 0.47 → 0.11 | 91 → 72 | 0.50 → 0.26 |
| V | BERT | PastFuture | S → V | 66 → 55 | 0.50 → 0.25 | 91 → 72 | 0.51 → 0.33 |

S = Surrogate model; V = Victim model

Table 5: **Cross-seed attack results.** In all cases, the efficacy of the white-box attacks (S→S) is significantly reduced while the efficacy of the transfer attacks (S→V) comparatively drops.

| Model | Arch. | Mechanism | Type | RTE | | QQP | |
|---|---|---|---|---|---|---|---|
| | | | | Acc. | Eff. | Acc. | Eff. |
| S | BERT | PastFuture | S → S | 66 → 52 | 0.47 → 0.11 | 91 → 72 | 0.50 → 0.26 |
| V | BERT | DeeBERT | S → V | 64 → 51 | 0.36 → 0.26 | 91 → 70 | 0.35 → 0.28 |
| S | BERT | PABEE | S → S | 66 → 49 | 0.22 → 0.08 | 91 → 69 | 0.35 → 0.16 |
| V | BERT | DeeBERT | S → V | 64 → 50 | 0.36 → 0.28 | 91 → 73 | 0.35 → 0.31 |
| S | BERT | DeeBERT | S → S | 67 → 55 | 0.32 → 0.11 | 91 → 76 | 0.32 → 0.18 |
| V | BERT | PABEE | S → V | 65 → 53 | 0.22 → 0.19 | 91 → 67 | 0.35 → 0.26 |
| S | BERT | PastFuture | S → S | 66 → 52 | 0.47 → 0.11 | 91 → 72 | 0.50 → 0.26 |
| V | BERT | PABEE | S → V | 65 → 57 | 0.22 → 0.18 | 91 → 76 | 0.35 → 0.31 |
| S | BERT | DeeBERT | S → S | 67 → 55 | 0.32 → 0.11 | 91 → 76 | 0.32 → 0.18 |
| V | BERT | PastFuture | S → V | 66 → 54 | 0.50 → 0.40 | 91 → 76 | 0.51 → 0.46 |
| S | BERT | PABEE | S → S | 66 → 49 | 0.22 → 0.08 | 91 → 69 | 0.35 → 0.16 |
| V | BERT | PastFuture | S → V | 66 → 55 | 0.50 → 0.29 | 91 → 74 | 0.51 → 0.38 |

S = Surrogate model; V = Victim model

Table 6: **Cross-mechanism attack results.** While not as effective as the cross-seed attack, marginal efficacy drops are seen for most victim-surrogate pairs.

# F More Discussion on Our Linguistic Analysis Results

Here, we provide further insights regarding our linguistic analysis performed in Sec 6. Conventional wisdom from studies in computer vision suggests that if an adversary leverages larger input perturbations (e.g., the perturbations are bounded to 16 pixels), their attack will be stronger than attacks with smaller input perturbations (e.g., 8 pixels). In other words, if a model is robust against attacks perturbing 16 pixels at most, the model is also robust to the 8-pixel bounded perturbations.

However, we find that this is *not true* for our slowdown attacks. Investigating the adversarial texts generated from our "unbounded" slowdown attacks, we could not find the correlation between the attack strength and the number of word-level perturbations. This result questions the effectiveness of adversarial training (AT), a standard defense that trains a model with bounded adversarial texts [26, 46, 14, 24, 43]. In Sec 7, we show that vanilla AT is an ineffective countermeasure (and also causes undesirable consequences, e.g., the utility and efficacy loss of a model).

We also offer an alternative insight for developing future defenses. In Sec 6, we show that an adversary can exploit the subject-predicate mismatch to make a model less confident about the perturbed sample's prediction. This misalignment, while easier for humans to identify, is difficult for a target model to do so. Thus, in Sec 7, we propose to leverage models able to correct grammatical errors, including the subject-predicate mismatches, for sanitizing inputs before being fed to the target

| Model | Arch. | Type | Metric | RTE | MNLI | MRPC | QNLI | QQP | SST-2 | CoLA |
|---|---|---|---|---|---|---|---|---|---|---|
| S | ALBERT | S → S | Acc.
Eff. | 77 → 55
0.25 → 0.12 | 85 → 30
0.28 → 0.06 | 88 → 49
0.33 → 0.11 | 90 → 55
0.32 → 0.07 | 91 → 71
0.36 → 0.19 | 92 → 56
0.36 → 0.07 | 81 → 49
0.32 → 0.08 |
| V | BERT | S → V | Acc.
Eff. | 65 → 51
0.22 → 0.20 | 83 → 24
0.24 → 0.16 | 82 → 43
0.34 → 0.10 | 89 → 69
0.29 → 0.24 | 91 → 78
0.35 → 0.34 | 91 → 71
0.32 → 0.21 | 82 → 59
0.34 → 0.24 |
| S | BERT | S → S | Acc.
Eff. | 66 → 49
22 → 0.08 | 82 → 30
0.24 → 0.06 | 81 → 54
0.33 → 0.14 | 89 → 57
0.29 → 0.08 | 91 → 69
0.35 → 0.16 | 91 → 55
0.32 → 0.07 | 77 → 63
0.37 → 0.26 |
| V | ALBERT | S → V | Acc.
Eff. | 77 → 63
0.22 → 0.21 | 85 → 22
0.28 → 0.16 | 86 → 81
0.32 → 0.22 | 91 → 74
0.32 → 0.22 | 91 → 74
0.36 → 0.34 | 92 → 68
0.36 → 0.28 | 81 → 47
0.22 → 0.22 |

S = Surrogate model; V = Victim model

Table 7: **Cross-architecture attack results.** With PABEE as the early-exit mechanism, we attack BERT-based models with an ALBERT-based surrogate and vice-versa on all GLUE tasks.

multi-exit models. But we find that such models are either far too slow to be practical or do not offer enough benefits. The result suggests future work in input sanitization for fast and effective methods.

## G  Impact of WAFFLE on Runtime

We provide results on the impact of our attacks on *runtime* and compare it with the efficacy metric we use. Table 8 shows our results on DeeBERT across multiple datasets.

| Metric | QQP | | RTE | | QNLI | | MRPC | | CoLA | |
|---|---|---|---|---|---|---|---|---|---|---|
| | Clean | WAFFLE | Clean | WAFFLE | Clean | WAFFLE | Clean | WAFFLE | Clean | WAFFLE |
| **Efficacy** | 0.36 | 0.22 | 0.34 | 0.12 | 0.35 | 0.09 | 0.35 | 0.10 | 0.34 | 0.13 |
| **Runtime** | 7.50s | 9.09s | 2.75s | 3.41s | 3.73s | 4.68s | 7.78s | 10.70s | 7.84s | 10.04s |

Table 8: **Impact of WAFFLE on runtime.** With DeeBERT as the victim's mechanism, we report the runtime (in seconds) and efficacy of clean and perturbed samples on five GLUE tasks. We run our experiments on a single Tesla V100 GPU. These results indicate that WAFFLE increases the actual runtime of multi-exit language models and that runtime is inversely correlated to efficacy.

The results show that WAFFLE increases the actual runtime of multi-exit language models, i.e., our slowdown results apply to real-world scenarios. Additionally, a reduction in efficacy is correlated with an increase in runtime. We choose efficacy as a metric to quantify the slowdown (as opposed to runtime) because the metric is hardware agnostic. The exit layer number we use to compute efficacy will not change between models run on different hardware configurations.

