# OpenReview forum: "BERT Lost Patience Won't Be Robust to Adversarial Slowdown"
_NeurIPS.cc/2023/Conference — NeurIPS 2023 poster_

### Official Review · Reviewer_69Mr · 2023-07-06

**Soundness:** 3 good
**Presentation:** 3 good
**Contribution:** 2 fair
**Rating:** 6
**Confidence:** 4

**Summary:**

The paper considers adversarial slowdown attacks on multi-exit text classification models based on BERT. They propose an attack, Waffle, which adapts text adversarial example attacks to a slowdown objective. They measure the susceptibility of multi-exit models for GLUE to their attack, evaluate attack transferability, analyze their generated adversarial text, and discuss mitigations for this attack.

**Strengths:**

The evaluation is quite broad in the classification tasks considered, multi-exit models used, and baselines. I also appreciated the transferability analysis and the "UAP".

I was interested to see the linguistic analysis of attacks. The linguistic markers mentioned seem plausible and I like the qualitative analysis in Table 3. However, I would like to also see some quantitative analysis of this property as well.

The paper is the first I am aware of to consider adversarial slowdown on text classifiers. This is a natural problem and may be of interest as text models become increasingly large.

**Weaknesses:**

Reading the paper, I was surprised that I never saw an experiment's running time measured, since this is the motivation for the attack. I think this is a pretty important consideration especially for defenses. If the running time of a defense (especially ChatGPT) is higher than the actual slowdown, there's not any point in applying the countermeasure (this is never discussed in the subsection).

The attack often creates incoherent text, as seen in Table 3. This seems to be a feature of text attacks, rather than a limitation specifically of the Waffle attack. However, it seems that the attack could be overfitting to a specific "distance metric" for the attack. Using some other distance functions, such as token/character edit distance, character replacement, as supported in TextAttack may also be useful to understand the generality of the attack.

ChatGPT is likely to have data leakage here, making it a bad scientific baseline. I would encourage at least a discussion of this. This may be one reason why the more standard grammar checking tools are unable to correct the attacks.

**Questions:**

What running time impact do your adversarial slowdown attacks have? How much time do the grammar checking countermeasures take?

How often do subject-predicate agreement and changing named entities result in slowdown? What fraction of successful slowdowns fall into these buckets?

**Limitations:**

The running time of experiments is never measured, which is a limitation that is not mentioned in the paper.

Data leakage in ChatGPT may also be a factor in the countermeasures.

---

> ### Author Rebuttal · Authors · 2023-08-09
>
> We thank the reviewer for the time to read and provide feedback. Below, we provide answers to your questions and concerns. We will also include them in the final version of our paper.
>
> —
>
> > (Question 1) What running time impact do your adversarial slowdown attacks have?
>
> We acknowledge the importance of measuring our attack’s impact on the actual runtime of multi-exit models. We first show our attack’s impact on the runtime in the results below. All samples are crafted using TextFooler with DeeBERT as the victim model (results are formatted as CLEAN &rarr; WAFFLE, CLEAN being the clean inputs and WAFFLE their perturbed counterparts):
>
> ---
>
> **QQP**
>
> Efficacy: 0.36 &rarr; 0.22
>
> Runtime: 7.5 s &rarr; 9.1 s
>
> **RTE**
>
> Efficacy: 0.34 &rarr; 0.12
>
> Runtime: 2.7 s &rarr; 3.4 s
>
> **MRPC**
>
> Efficacy: 0.35 &rarr; 0.09
>
> Runtime: 3.7 s &rarr; 4.7 s
>
> **QNLI**
>
> Efficacy: 0.35 &rarr; 0.10
>
> Runtime: 7.8 s &rarr; 10.7 s
>
> **CoLA**
>
> Efficacy: 0.34 &rarr; 0.13
>
> Runtime: 7.8 s &rarr; 10.0 s
>
> ---
>
> As is evident, a reduction in efficacy is correlated with an increase in runtime. Our usage of efficacy in our paper mainly comes down to it being hardware agnostic, as the exit layer will not change between models run on different machines. This makes it a strong metric for quantifying the speed-up of multi-exit models, regardless of the exact experimental setup.
>
> —
>
> > (Question 2) How much time do the grammar checking countermeasures take?
>
> The runtime of the grammar-checking countermeasures explored in Sec. 7 is higher than the inference time of all multi-exit models our work considers. As the reviewer accurately points out, this would make them useless from a pure runtime standpoint. However, we reason that the purpose of using these defenses was primarily exploratory, aiming to understand further why specific text causes more slowdown and how modifying such text can revert this slowdown. Moreover, input sanitization is already used in commercial models. Claude-2 [1], a conversational model similar to ChatGPT, already employs input-filtering techniques, which we believe is a promising future work direction.
>
> We acknowledge the importance of defense mechanisms with less computational costs and consider this to be an important area for future work. We will include this discussion in Appendix E.
>
> [1] https://claude.ai/
>
> —
>
> > (Concern 1) The attack often creates incoherent text, as seen in Table 3. This seems to be a feature of text attacks, rather than a limitation specifically of the Waffle attack. However, it seems that the attack could be overfitting to a specific "distance metric" for the attack. Using some other distance functions, such as token/character edit distance, character replacement, as supported in TextAttack may also be useful to understand the generality of the attack.
>
> We acknowledge that incoherent texts are a limitation of adversarial attacks in the natural language processing domain, and WAFFLE is not exempt from this limitation. The specific distance metric used for WAFFLE depends on its underlying adversarial crafting algorithm; in the case of our experiments, we use the distance metrics in-line with the original work of [1] and [2]. We clarify that WAFFLE itself has no distance metric, instead providing an objective function that adversarial crafting algorithms may utilize with their own respective distance metrics. We leave the exploration of different distance metrics as future work, as well as exploring newer adversarial crafting algorithms that boast greater sentence coherence (e.g. [3]).
>
> [1] Jin et al., Is BERT Really Robust? A Strong Baseline for Natural Language Attack on Text Classification and Entailment, arXiv 2019
>
> [2] Yoo & Qi, Towards Improving Adversarial Training of NLP Models, ACL 2021
>
> [3] Li et al., Contextualized Perturbation for Textual Adversarial Attack, ACL 2021
>
> —
>
> > (Question 3) How often do subject-predicate agreement and changing named entities result in slowdown? What fraction of successful slowdowns fall into these buckets?
>
> To stress the prevalence of subject-predicate disagreement and changing of named entities in samples crafted by WAFFLE, we follow the reviewer's advice and conduct an experiment that categorizes such samples into "buckets." Taking the top-100 slowdown-inducing adversarial texts from QQP crafted on DeeBERT, we quantify the number that contain at least one instance of subject-predicate disagreement or at least one instance of a changed named entity. 84% introduce some form of subject-predicate disagreement, and 31% change a named entity. This result is consistent with our linguistic analysis and speaks to BERT's importance on these factors when performing inference. An important consideration is that not all samples contained a named entity, explaining why the prevalence of named entity changes was much lower despite consistently contributing to the slowdown. We thank the reviewer for considering this quantitative approach and will update our final paper to include this result.
>
> —
>
> > (Question 4) ChatGPT is likely to have data leakage here, making it a bad scientific baseline. I would encourage at least a discussion of this. This may be one reason why the more standard grammar checking tools are unable to correct the attacks.
>
> We thank the reviewer for bringing this to our attention. We acknowledge the likelihood of ChatGPT leaking data and will discuss the risks in our limitations and societal impact section (Appendix E). We did not mean to suggest ChatGPT is a practical defense (execution time aside). Instead, since other input sanitation (Grammarly) failed, we used it as an accessible tool for input sanitization via a conversational model as a proof-of-concept that it may have effectiveness as a defense. We envision future work on evaluating the robustness and efficiency of input sanitization defenses using conversational models (with better controlled and known datasets).

---

> > ### Comment · Reviewer_69Mr · 2023-08-11
> > **Thank you**
> >
> > I'm happy with the response and will increase my score.

---

> > > ### Author Response · Authors · 2023-08-11
> > > **Thank You**
> > >
> > > Dear Reviewer 69Mr, We would like to thank you again for taking the time to read our rebuttal. We are happy that our response addresses your concerns and questions. We will make sure our responses are reflected in the final version of our paper.

---

### Official Review · Reviewer_Lh1p · 2023-07-06

**Soundness:** 3 good
**Presentation:** 3 good
**Contribution:** 3 good
**Rating:** 7
**Confidence:** 2

**Summary:**

This papper proposes WAFFLE, a slowdown attack to generate natural adversarial text bypassing early-exits.
Empirical results show the robustness of multi-exit language models against adversarial slowdown.

**Strengths:**


1.The paper is well-written and easy to follow.

2.The evaluation is comprehensive.


**Weaknesses:**


1.It seems reference[1] does similar slowdown attacks, but [1] works on computer vision domain. What are the differences between WAFFLE and [1]?

2.Does WAFFLE still work on non-transformer based architectures, such as LSTM?



[1] Hong, S., Kaya, Y., Modoranu, I.V. and Dumitras, T., 2020, October. A Panda? No, It's a Sloth: Slowdown Attacks on Adaptive Multi-Exit Neural Network Inference. In International Conference on Learning Representations.


**Questions:**

1.Whether the proposed method can still work properly on other transformer architectures, such as GPT-2, RoBERTA?


**Limitations:**

The authors adequately addressed the limitations.

---

> ### Author Rebuttal · Authors · 2023-08-09
>
> We thank the reviewer for the time to read and provide constructive feedback. Below, we answer the questions and concerns. We will also include this discussion in the final version of our paper.
>
> —
>
> > (Weakness 1) It seems reference [1] does similar slowdown attacks, but [1] works on computer vision domain. What are the differences between WAFFLE and [1]?
>
> We first clarify that while our work and the work done in [1] both achieve slowdown against multi-exit models, differences in problem domain lead to differences in approach. (1) Against language models, we often do not have access to input gradients (which is straightforward in attacks against computer-vision models). We thus need to design a new slowdown objective compatible with non-gradient-based attacks. (2) We must bound the values of our slowdown objective within [0, 1]. We found that the objective used in the prior work [1] is unbounded to [0, inf]; thus, a straightforward adaptation of this objective for adversarial text-attack algorithms leads to unbounded perturbations, and the resulting text completely differs from the original one. (3) The attack against language models works with discrete text inputs; not all embedding-level perturbations we compute exist as words and small changes to input (word, characters) can result in large logit changes. We must search for candidate words (or word combinations) for substitution. Due to the space limits, we summarized the challenges in Line 74–78, but for clarity, we will include this discussion in the final version of our paper.
>
> [1] Hong et al., A Panda? No, It’s a Sloth: Slowdown Attacks on Adaptive Multi-exit NN Inference
>
> —
>
> > (Weakness 2) Does WAFFLE still work on non-transformer based architectures, such as LSTM?
>
> Given the nature of our attack, it is necessary that a victim model contain early-exits. To the best of our knowledge, there is no prominent work on LSTM based early-exit models for the natural language processing domain. So, we have not tested on LSTM models. We focus on transformer-based architectures due to their effectiveness and future potential in the NLP domain. However, if there is a recommended LSTM model, we would be happy to investigate.
>
> —
>
> > (Question 1) Whether the proposed method can still work properly on other transformer architectures, such as GPT-2, RoBERTA?
>
> We provide evidence of WAFFLE’s transferability between different models and architectures by testing three different model-architecture pairs in our main analysis in Sec. 4. Further combinations are tested in Sec. 5, which shows that WAFFLE transfers well between transformer-based models.
>
> To answer the reviewer's question explicitly, we tested several multi-exit models with RoBERTa and found similar slowdown results as the BERT versions. This is unsurprising because RoBERTa is essentially a BERT model but with a modified and improved set of hyperparameters. At the time of our implementation, GPT-like early exit models were not yet prominent and it would have taken significant time and resources to modify GPT-2 with early exit methods and retrain the models. We acknowledge that some now exist, such as [1, 2], and results would be interesting to see.
>
> [1] Schuster et al., Confident Adaptive Language Modeling, NeurIPS 2022
>
> [2] Din et al., Jump to Conclusions: Short-Cutting Transformers With Linear Transformations, arXiv 2023

---

> > ### Comment · Reviewer_Lh1p · 2023-08-14
> > **Thanks for explaination.**
> >
> > Thanks for the detailed explaination. The reviewer is satisfied with the response.

---

### Official Review · Reviewer_9ThA · 2023-07-08

**Soundness:** 3 good
**Presentation:** 3 good
**Contribution:** 3 good
**Rating:** 5
**Confidence:** 4

**Summary:**

The paper evaluates robustness of multi-exit language models against specifically perturbed datapoints that induce adversarial slowdown and controbutes to the literature on availability attacks. The paper targets language models, as opposed to the vision models that were explored in the existing literature. The paper presents WAFFLE and attack that forces the early exists to be avoided and ultimately slows down the computation. The paper then explores the constracted examples and finds that they appear slightly more out of distribution.

**Strengths:**

+ Interesting important setting

**Weaknesses:**

+ Unclear performance with respect to the related work

**Questions:**

Thank you very much for the paper, I enjoyed the read very much. I really only have two comments.

1. To the best of my knowledge, availability attacks against language models were previously done by Shumailov et al. with Sponge examples (EuroS&P, https://arxiv.org/abs/2006.03463) and similarly adopted by Boucher et al (reference [1] in the paper). How would these attacks do in comparison to WAFFLE?

2. Would you expect to see more performance degradation if text were turned even more out of distribution? Are there cases where ChatGPT defence would not work?

**Limitations:**

-

---

> ### Author Rebuttal · Authors · 2023-08-09
>
> We thank the reviewer for the time to read and provide feedback. Below, we provide answers to the questions and concerns. We will also include this discussion in the final version of our paper.
>
> —
>
> > (Question 1) To the best of my knowledge, availability attacks against language models were previously done by Shumailov et al. with Sponge examples (EuroS&P) and similarly adopted by Boucher et al (reference [1] in the paper). How would these attacks do in comparison to WAFFLE?
>
> We first clarify that our attack is the first of its kind: a slowdown attack that generates natural adversarial text that bypasses the early-exit layers of multi-exit models. Sponge examples [1] (and as used in  Bad Characters [2]) are ‘resource’ attacks. They exploit computational properties of hardware or tokenization, e.g., input dimensionality and/or activation sparsity, to increase the inference runtime. In contrast, our attack is hardware-agnostic and targets multi-exit model architectures, a new algorithm for efficient language model computations.
>
> [1] Shumailov et al., Sponge Examples: Energy-Latency Attacks on Neural Networks, IEEE 2021
>
> [2] Boucher et al., Bad characters: Imperceptible nlp attacks, IEEE 2022
>
> —
>
> > (Question 2) Would you expect to see more performance degradation if text were turned even more out of distribution? Are there cases where ChatGPT defense would not work?
>
> Following our linguistic analysis in Sec. 6, we do suspect that further pushing samples towards out-of-distribution would further degrade performance. The quantification of an out-of-distribution sample may be non-trivial, but we believe some evidence of the previous claim is exhibited in Fig. 2. As we increase the attack success threshold, i.e. the score needed to satisfy our slowdown objective in Sec. 3.2, both the accuracy and efficacy of the victim models decrease. This could in part be due to these samples being pushed further out-of-distribution, a claim in alignment with our linguistic analysis.
>
> We also agree that it is an interesting question to ask to what extent conversational models like ChatGPT offer robustness to adversarial slowdown. However, it is not the scope of our work and requires future work. While ChatGPT shows some robustness to OOD and adversarial attacks in the prior work [2], we clarify that our claim is **not** that ChatGPT is robust to any attacks with input text perturbations. Instead, as a tool for realizing input sanitization, we observe that ChatGPT provides some effectiveness with fewer side-effects. We next envision future work on evaluating the robustness of input sanitization via conversational models against adaptive adversaries; recent work [3] would be a nice starting point. We will include this discussion in the final version of our paper.
>
> [2] Wang et al., On the Robustness of ChatGPT: An Adversarial and Out-of-distribution Perspective
>
> [3] Zou et al., Universal and Transferable Adversarial Attacks on Aligned Language Models

---

> > ### Comment · Reviewer_9ThA · 2023-08-12
> >
> > Many thanks for your response.
> >
> > I am not sure I agree with the hardware-specifics discussions above and the distinction. One can think about auto-regressive models as a type of early-exit, where smaller sequences use less compute -- Sponge examples with their focus on larger input/output sequences cause the exit strategy to change for them.  Never the less, as long as the discussion of the differences is included into the final draft I am happy with bumping the score.

---

> > > ### Author Response · Authors · 2023-08-13
> > > **Thank You**
> > >
> > > We would like to thank the reviewer again for taking the time to read our rebuttal. We agree, in particular for the case of auto-regressive models, there may be similarities to sponge examples. We will make sure to update the paper with a discussion on sponge examples.

---

### Official Review · Reviewer_KooQ · 2023-07-08

**Soundness:** 3 good
**Presentation:** 3 good
**Contribution:** 3 good
**Rating:** 6
**Confidence:** 4

**Summary:**

This paper evaluates the robustness of multi-exit language models against adversarial slowdown. The authors propose a slowdown attack that generates natural adversarial text to bypass early-exit points. They conduct a comprehensive evaluation of three multi-exit mechanisms using the GLUE benchmark and demonstrate that their attack significantly reduces the computational savings provided by these mechanisms in both white-box and black-box settings. The study reveals that more complex mechanisms are more vulnerable to adversarial slowdown. Adversarial training is found to be ineffective in countering the slowdown attack, while input sanitization with a conversational model like ChatGPT can effectively remove perturbations. The paper concludes by emphasizing the need for future research in developing efficient and robust multi-exit models.

**Strengths:**

- Proposed the first slow-down attack on language models.
- Evaluate the proposed methods on different architectures (i.e., early-exit mechanisms)
- Demonstrate the effectiveness of the methods in different threat models/attack settings (i.e., black-box, white-box)
- Analysis on generated adversarial examples is conducted to provide further insights into the vulnerability of the model
- Mitigation and defense methods are discussed to show that sanitization methods are more effective compared to robust training method.

**Weaknesses:**

- Since the slow-down attack has been demonstrated in vision task, the challenge of adapting it to language model is not clear.
- The proposed slow-down objective seems trivial in terms of novelty.

**Questions:**

In general, this paper is well written with complete story and comprehensive analysis. The reviewers would appreciate if the authors could improve on several perspectives:
- Highlighting the difference of slow-down attack between language and vision task to show the challenges this paper resolved.
- Providing some discussing regarding the linguistic analysis on the adversarial examples to provide actual insights of improve the language model/exit classifiers in the future.
- Limitation of the paper should be discussed to shed light on potential future works along this direction.

**Limitations:**

The authors did not have a section regarding the limitation discussion.

---

> ### Author Rebuttal · Authors · 2023-08-09
>
> We thank the reviewer for the time to read and provide valuable feedback. Below, we provide answers to the questions and concerns. We will also include this discussion in the final version of our paper.
>
> —
>
> > (Weakness 1 and Question 1) Highlighting the difference of slow-down attack between language and vision task to show the challenges this paper resolved.
>
> We first clarify the unique challenges we addressed in developing our slowdown attacks, compared to ones developed for computer-vision models, as follows: (1) Against language models, we often do not have access to input gradients (which is straightforward in attacks against computer-vision models). We thus need to design a new slowdown objective compatible with non-gradient-based attacks. (2) We must bound the values of our slowdown objective within [0, 1]. We found that the objective used in the prior work [1] is unbounded to [0, inf]; thus, a straightforward adaptation of this objective for adversarial text-attack algorithms leads to unbounded perturbations, and the resulting text completely differs from the original one. (3) The attack against language models works with discrete text inputs; not all embedding-level perturbations we compute exist as words and small changes to input (word, characters) can result in large logit changes. We must search for candidate words (or word combinations) for substitution. Due to the space limits, we summarized the challenges in Line 74–78, but for clarity, we will include this discussion in the final version of our paper.
>
> [1] Hong et al., A Panda? No, It’s a Sloth: Slowdown Attacks on Adaptive Multi-exit NN Inference
>
>
> > (Weakness 2) The proposed slow-down objective seems trivial in terms of novelty.
>
> Thank you for your comment, we acknowledge that we could have been more clear on the novelty of our work and will expand the Appendix accordingly.  As we clarified in our response to the first question, due to the differences between CV and NLP, it was unclear how well the attacks would work if at all.  For space and readability reasons, we did not show all of the variations of the attack, search methods, and objective functions that we attempted in order to find the most effective methodology. Additionally, NLP has additional dimensions of semantics and sentence structure which we analyze in order to understand why the attacks work and how to defend against them.
>
> —
>
> > (Question 2) Providing some discussing regarding the linguistic analysis on the adversarial examples to provide actual insights of improve the language model/exit classifiers in the future.
>
> We thank the reviewer for pointing out this. We acknowledge that due to the space limitations, our connections to future work are unclear. We first clarify that our purpose of providing the linguistic analysis (Sec 6) is not just to provide our attack’s characteristics but to shed light on future directions for bringing efficient and robust multi-exit language models. We highlight some of the less clear  lessons Sec. 6 offers as follows, and we will also update the final version of our paper to better describe them.
>
> **Models Robust to Larger Perturbations May Not Be Robust to Smaller Perturbations**
>
> Conventional wisdom from studies in computer vision is that: if an adversary leverages larger input perturbations (e.g., the perturbations are bounded to 16 pixels), their attack will be stronger than the attacks with smaller input perturbations (e.g., 8 pixels). In other words, if we robustify a model against the attacks perturbing 16 pixels at most, the model is also robust to the 8-pixel bounded perturbations.
>
> However, we show this is not true for our slowdown attacks. Investigating the adversarial texts generated from our “unbounded” slowdown attacks, we could not find the correlation between the attack strengths and the perturbation amounts. It questions the effectiveness of adversarial training, a conventional defense that trains a model with bounded adversarial texts. The observation led to our first experiments on potential countermeasures, and we show that adversarial training is ineffective (and also causes undesirable consequences, e.g., the utility and efficacy loss of a model). For future work this suggests adversarial training is still not mature and may need to utilize linguistic information.
>
> **Input Sanitization May Be A Promising Direction for Defeating Slowdown Attacks**
>
> Sec 6 offers an alternative insight for developing future defenses. We show that an adversary can exploit the subject-predicate mismatch to make a model less confident about the perturbed sample's prediction. This misalignment, while easier for humans to identify, is difficult for a target model to do so. Thus, in Sec 7, we propose to leverage models able to correct grammar errors, including the subject-predicate mismatches, for sanitizing inputs before being fed to the target multi-exit models. However, input sanitization may be slow which offsets the early exit speedup and we showed that some sanitization methods may not be effective either. This suggests future work in input sanitization for fast and effective methods, where we uncovered some key linguistic features that may be necessary.
>
> —
>
> > (Question 3) Limitation of the paper should be discussed to shed light on potential future works along this direction. The authors did not have a section regarding the limitation discussion.
>
> We kindly remind the reviewer that we discuss the limitations and future work in Appendix E.

---

> > ### Comment · Reviewer_KooQ · 2023-08-11
> >
> > Thanks for the detailed explaination/clarifications. The reviewer is satisfied with the response besides a few comments:
> >
> > > Models Robust to Larger Perturbations May Not Be Robust to Smaller Perturbations
> >
> > This seems to be counter-intuitive. Usually a large perturbation is a superset for smaller perturbation, right? Did the reviewer miss anything for this insights?
> >
> > > Limiatation sections
> >
> > It would always be a good practice to have a reference in the main paper for contents in appendix, especially for important contents like discussions of limitations :)

---

> > > ### Author Response · Authors · 2023-08-11
> > > **Thank You and Our Response to Additional Questions**
> > >
> > > Dear Reviewer KooQ, We first would like to thank you again for taking the time to read our rebuttal.
> > >
> > > > This seems to be counter-intuitive. Usually a large perturbation is a superset for smaller perturbation, right? Did the reviewer miss anything for this insights?
> > >
> > > We clarify that the perturbation is the “word-level” perturbations, i.e., how many “words” are perturbed in crafting adversarial texts. This is different from “numerical perturbations” to the inputs that attacks against computer vision models use. We observe from our experiments in Sec 6 that it is important for the attacker to choose the “right” word(s) to cause slowdowns rather than perturbing many words.
> > >
> > > We also point out that existing adversarial text attacks, e.g., [1, 2], correlate (or quantify) the attack strengths as the % of perturbed “words.” Nevertheless, our work highlights that it may not be the right metric, as even a single word can be sufficient to craft a strong adversarial text.
> > >
> > > > Limitation Sections
> > >
> > > Thank the reviewer for a great suggestion. We make sure to have references to the Appendix in the main paper.
> > >
> > > ---
> > >
> > > We are happy to answer any further questions (or concerns); let us know.
> > >
> > > [1] Jin et al., Is BERT Really Robust? A Strong Baseline for Natural Language Attack on Text Classification and Entailment.
> > > [2] Yoo et al., Towards Improving Adversarial Training of NLP Models

---

> > > > ### Comment · Reviewer_KooQ · 2023-08-21
> > > >
> > > > Thanks for the explanation. I am satisfied with the response.

---

### Decision · Program_Chairs · 2023-09-21

**Decision:**

Accept (poster)

**Comment:**

Summary of the reviews:

The paper proposes a slowdown attack, WAFFLE, against multi-exit language models. The attack is evaluated on three multi-exit mechanisms and shown to be effective in both white-box and black-box settings. The paper also analyzes the generated adversarial examples and finds that they are often linguistically plausible. Finally, the paper discusses potential mitigations for the slowdown attack, including adversarial training and input sanitization.

Strengths:
- The paper is well-written and the evaluation is comprehensive.
- The paper provides insights into the vulnerability of multi-exit language models.

Weaknesses:
- There are some concerns from reviewers about the novelty of the slow-down attack, which was demonstrated in vision tasks.

Overall, the four reviews are generally positive. All of them acknowledge the importance of the problem that the paper addresses, and most of them find the proposed solution to be effective. I would recommend this paper for acceptance.